# Changing Health Behavior with Social Technology? A Pilot Test of a Mobile App Designed for Social Support of Physical Activity

**DOI:** 10.3390/ijerph17228383

**Published:** 2020-11-12

**Authors:** Anne Marie Kanstrup, Pernille Scholdan Bertelsen, Casper Knudsen

**Affiliations:** Department of Planning, Aalborg University, 9000 Aalborg, Denmark; pernille@plan.aau.dk (P.S.B.); ckn@plan.aau.dk (C.K.)

**Keywords:** mobile health, social support, physical activity, participatory design

## Abstract

Mobile applications targeting people engaged in physical activity have increased. However, while research has identified social support as a key factor for people’s engagement in physical activity, most mobile health (mHealth) applications are designed for individual use. In this paper, we report on a research study exploring opportunities for designing mHealth to facilitate social support around physical activity. A mHealth application was designed, and pilot tested for eight weeks with healthcare professionals (n = 3) and two groups of citizens (n = 20) who were motivated but challenged physically due to various health conditions. Data was collected via online monitoring of the use of the mHealth application during the pilot test and via qualitative interviews with the participants before and after. The results support the idea that designing for social health support is important but so is identifying key challenges related to (i) the facilitation of technology-mediated social health support, especially to a target group that is living with health challenges, and (ii) finding a balance between social and health agendas that bring social support to the foreground for the participants.

## 1. Introduction

Physical inactivity is a leading risk factor for mortality [1], and guidelines for weekly physical activity [2] are difficult for many people to meet, especially those who are not socially connected. A study of 3342 adults in six European countries identified that people with limited social support from family, friends, school, and work were more than twice as likely to be inactive than people with high social support [3]. Social support is recognized as a key factor for motivating people to engage in physical activity and improving health in general [4]. Though there are complicated mechanisms that contribute to health benefits from social support, research has shown that cooperation among people in similar situations has a strong effect on their ability to change health habits [5]. Moreover, well-being is highest among people who have a perceived experience of available social support [6]. For this reason, it is relevant to research how to advance social support among people aiming to engage in physical activity and a healthy lifestyle, and while doing so, pay special attention to how to facilitate healthy living cooperation between people on an everyday basis, and how to provide an increased experience of available social support for a healthy lifestyle.

Digital technology has introduced new opportunities for mediating social support for everyday health management. For example, recent research shows how various types of online support systems can facilitate social support for weight loss [7], nutrition [8], and management of chronic illnesses [9], as well as how social technology can be designed to mediate social relations among users in a local community to support each other in daily health issues [10]. Mobile health applications (mHealth) designed to support people in physical activity have increased rapidly alongside smartphone users overall increase [11]. A recent review of 165,000 consumer health applications available via the Apple App Store or Google Play Store showed that two-thirds of mHealth apps focus on everyday health promotion, including physical activity, diet, stress, and lifestyle [12]. However, most mHealth applications are designed for individual use, with limited attention to facilitating social support [13]. 

In this paper, we present results from a long-term research project where we have designed and piloted a mHealth application intended to mediate social support among people who are motivated but challenged to be physically active due to various health and life conditions. Our overall research question focuses on understanding how mobile technology can provide social support for healthy lifestyles among physically inactive people. Hence, the presented research contributes insights on how to complement the design and use of mHealth applications primarily focused on individual use with social features and technology-mediated support. This is done to provide functional assistance to people in need of social health support. 

## 2. Methods

### 2.1. A Participatory Design Process in a High-Risk Health Zone

The research presented in this paper is part of a long-term participatory design research exploring opportunities and challenges for decreasing inequality in digital health [14]. The increase of digital technology designed and used in the delivery of health has raised concerns that most technologies designed for health come with a risk of reinforcing health inequality since they are designed predominantly for “people like us” (PLUs), defined as “people who believe they understand healthcare and health issues, take care of their own health, are literate, well-to-do, tech-savvy [15].” A consequence of this is that mHealth program designers miss important target groups if they fail to engage with and design for people living with health and life challenges, i.e., the people in greatest need of comprehensive health support. Against this background, we have set up our study as participatory design research with citizens who are motivated but challenged in doing physical activity. Participatory design is a discipline that offers methods for engaging multiple voices in technology design [16], including people who are marginalized by a somewhat elite-oriented tech industry [17].

Our research takes place in Denmark, where social health inequality is high (measured on mortality), which is significant since Denmark has one of the world’s smallest income gaps and a welfare system that is based on free access to education and healthcare. Studies show that social inequality based on mortality and health has doubled in Denmark over the past 25 years, and recent mappings of Danish citizens’ health (measured through food, smoking, alcohol, exercise, and mental health) confirm that health inequality is concentrated in residential areas dominated by citizens with low (or no) education beyond primary school and high unemployment. In these areas, a large percentage of the residents smoke, are obese and have low physical activity. Furthermore, many residents suffer from chronic illnesses and mental health problems compared to other residential areas [18]. The mapping of health inequality in Denmark corresponds with international literature that identifies a clear relationship between residential neighborhoods and health [19]. Against this background, we started our research by identifying a neighborhood that the national health authorities had highlighted as a high-risk health zone and established cooperation with citizens, healthcare professionals, and community workers in the area. The neighborhood selected as the field for our research is about 12 square kilometers, and approximately 16,000 residents live there. We cooperated primarily with residents in the social housing area, which accounts for 60% of the neighborhood. Since 2013, we have:Conducted participatory design walks with residents, healthcare professionals, and community workers to learn about health in the neighborhood [14].Staged participatory design interventions with residents living with depression, anxiety, and a low level of physical activity, resulting in the design requirements for mHealth support [20].Developed the first digital prototype and carried out a usability test with many residents, resulting in improved design and navigation [21].Conducted workshops with groups of residents and healthcare workers to learn how to bring the app into use among residents in the neighborhood [22].Launched the second version of the mobile app in the App Store and Google Play and staged pilots exploring the app’s use among residents signed up for health programs in the neighborhood.

This paper presents results from the pilots of the mHealth app.

### 2.2. The mHealth App

The literature on social support and health has highlighted the importance of realizing that social support is available [6] and having opportunities to access and participate in cooperation around health [5]. The empirical data from our participatory design process supported these core elements—participants in the study identified the importance of seeing activities accessible in their neighborhood and people interested in doing those activities. As expressed by one of the participants, “Where are people and what are they doing? If I could see this, I would consider joining activities, but I cannot see anything.” This statement spearheaded what became the conceptual model for digital design. We coined the importance of designing for a ‘healthy horizon,’ pointing toward a need to use digital technologies as an opportunity to provide transparency and share information in an easily accessible way via mobile apps. It became manifested in a design that provided the user with simple access to browse exercise activities in their residential area [22]. A second key finding from the participatory design process was that exercise was not the participants’ major challenge. It was, in fact, surrounding issues, like being able to plan for the activity, finding the mental resources (instead of excuses) to leave the house and join activities, being insecure, etc. (cf. [20] for elaboration). Against this background, we identified the importance of designing easy ways to join activities and become part of a ‘healthy cooperation’ with other people in the neighborhood. This healthy cooperation manifested itself in the simple functionality of creating and/or joining groups and thereby coming together for physical activity.

On this basis, a mobile app was designed around a user profile in which the selection of the residential area/ZIP code links residents’ profiles to other residents and activities in their local area. To provide flexibility for broader networks, more than one ZIP code can be chosen. Users can choose preferred exercise activities and filter activities depending on their own interests. The entry screen lists activities in the local area selected by the user profile. From this entry screen, users browse through available activities (the healthy horizon) and can, for each activity, choose to sign up, show interest in, or simply skip and browse to the next activity. To participate in an activity, the user taps ‘participate’ (‘*Deltager*’ in Danish cf. Figure 1, left). The user’s profile photo is then added to the activity for feedback and transparency (i.e., so others can see who is joining an activity) (Figure 1, left). If the user taps other participants’ photos, a new screen with a detailed list of participants’ names opens up (Figure 1, right). Activities are listed per day. Completed activities are shown until midnight but are marked in red and accompanied by the text ‘the activity is complete.’ In the lower part of the app, users can navigate the core functionalities of ‘activities,’ ‘groups,’ ‘notifications,’ and ‘profile’ (Figure 1, left). ‘Groups’ is a functionality that allows users to form groups and join groups, and through this, they create specific and private activities—groups can be private or public depending on user choice. When users tap ‘notifications’ (the bell icon), a list of received notifications appears. Profile settings (including preferred types of activities) can be changed by tapping ‘profile.’

In the upper-right corner of the entry screen (Figure 1, left), the resident can tap the ‘plus’ sign and create activities within selected types of exercise. This opportunity is fundamental to the design, as it supports a bottom-up approach to creating healthy activities and forming a healthy horizon among residents in a local area. When creating an activity, the user chooses a name for the activity, types in a location for where the participants should meet, selects a start time and expected duration, and decides who may participate, as an activity can be either public or for invited people only. Users who sign up for activities receive a notification 30 min before each activity starts or if activities are canceled or changed. The app also has a series of screens in a deep hierarchy that users can tap to find details about activities and groups, plus pop-ups designed to provide feedback and manage groups and activities. Additionally, it is possible to create a series of events in a simple manner [21].

### 2.3. Pilot Test

A pilot test of the mobile app was set up in cooperation with the neighborhood’s health center where the research was conducted. Following the ambition to design and pilot a mHealth application aimed to mediate social support among people who are motivated but feel challenged to be physically active, two target groups were selected for the pilot test:(i)Pilot A: A group of citizens living in the neighborhood (a zone of 16 km^2^). This group met once per week, facilitated by a community worker from the local health center. Activities had a health purpose but were organized informally, meaning that participants could also suggest activities. For example, walking tours were combined with collecting stones one week, while the next week, the group could decide to stay indoors and decorate/paint the stones. Everyone in the neighborhood could enroll in the group and participate in the activities. At the time, all Pilot A participants faced severe health issues (especially chronic illness and mental illness). Pilot A was conducted over eight weeks following planned activities in the group. A public group with their name was made in the mHealth application, meaning everyone in the neighborhood could enroll in their activities.(ii)Pilot B: A group of citizens who were geographically distributed all over the municipality (a zone of 1140 km^2^). This group also met once per week, and their activities were facilitated by two healthcare professionals from a municipal health center. Their activities followed a formal program for health promotion. For example, some activities focused on exercise and others on diet. Participants were referred to this group by their general practitioners (GPs). The participants had multiple different health challenges. Pilot B was conducted over five weeks following the planned activities in the group. For Pilot B, a private group was set up, meaning that activities were only visible and accessible to the group members.

Table 1 presents the composition of participants in the pilot tests. Four qualitative group interviews were conducted: one interview with each group before starting their pilot test and one interview with each group after ending their pilot test. Additionally, qualitative interviews were conducted with the three group facilitators—the community worker who participated in Pilot A and the two healthcare professionals who participated in Pilot B. In total, 23 participants were interviewed. During the intervention, we monitored activities in the application daily. Data were collected via screen dumps of activities in the application and manual registration of the number of participants.

Table 2 presents an overview of the activities and data conducted during the pilot test. As outlined, the pilot started with an introductory conversation with the facilitators agreeing on a mode for introducing the app to participants and instructing them in its use. Following this, a group for each pilot was created in the app in cooperation with the facilitators before the intervention. This ensured initial content in the app when the users started. At the introduction, participants were introduced to the app as a newly integrated part of their regular activities. We facilitated this by participating in the first meeting/activity and providing technical support for installing the app when needed. Facilitators and participants were encouraged to create activities themselves and use the application, i.e., to insert content into the app and use it for social support, and, in general, to establish a workflow around the app.

A qualitative analysis was conducted of the data material. Data were sorted into the initial overall themes to be explored during the pilot: ‘healthy horizon’ (the mHealth app’s ability to provide a sense of perceived available social support) and ‘healthy cooperation’ (the mHealth app’s ability to support cooperation between users during exercise). First, online data were analyzed to create a baseline understanding of the app’s use among the participants during the pilots. Second, interview data were transcribed and sorted into the two themes; after that, a re-reading of each group of data material was done to identify prevalent patterns. The analysis presents participants’ experiences using the app, experiences with the design for a healthy horizon, and experiences with healthy cooperation. On this basis, we synthesized the analytic results and identified two key findings, as presented in the following.

## 3. Results

### 3.1. Digital Activities—Use of the App During the Intervention

Online monitoring of the application’s use during the pilot revealed different types of use by each group, but generally limited use of the application by all participants.

In Pilot A, the facilitator quickly established the practice of appropriating the group’s activities to the mobile app. Specifically, the facilitator inserted the group’s activities into the app (mainly walking tours) with descriptions that reflected developing plans for walks in the near future as exemplified in Figure 2. Thus, participants could get a detailed overview of upcoming activities via the app, creating a horizon as envisioned for the design. However, only a few participants signed up for these activities, while several used the app to send cancellations for scheduled events. Pilot A participants also established a mode for using the app to create an overview of group activities. Each week, approximately half the group signed on as either participating or not participating; the number of participants grew slowly throughout the test to a total of nine users. Approximately half the participants (4–5 per activity) signed up for attendance the day before or the day of each event. The Pilot A group continued using the app after the intervention period. Some participants (n = 4) decided to use the mHealth application as a tool to organize a parallel walking group. They effectively took over the role of facilitator and created new activities for themselves every month. This adaption of the app is important since it indicates a clear feasibility for the app’s target purpose. It was, however, limited to the group already established within the app—none of the participants responded to activities outside the group, and no newcomers signed up for the group’s activities during the intervention, though all activities were publicly announced. 

In contrast, participants in Pilot B did not use the app during the intervention. The group activities (the course program) were mirrored and created as activities in the app by the facilitators, and all participants were invited via the app for each course session and additionally for a walk after each session. However, none of the participants used the app to show their attendance or otherwise engaged with any activities after joining the mHealth group. They only participated physically in the course throughout the test period. At the concluding interview, it became clear that participants had not been introduced to the app by the course facilitators as part of the orientation to the course. The course facilitators also made minimal use of the group function. One of the course facilitators engaged herself once by changing the description of an activity and by changing her status to ‘participating’ on some of the activities but otherwise showed no systematic or consistent use.

At the concluding interview with Pilot B participants, respondents mostly talked about their expectations of the mHealth app’s activity feed and reported that they were disappointed that the list of available activities was limited, i.e., that they experienced a lack of critical mass—a limited horizon—in the app during the intervention. It is important to compare Pilot B’s usage of the app with that of Pilot A. This indicates that the role of the facilitator(s) is important for engaging participants in the use of the app, not least by linking this use to actual participation in activities. Additionally (cf. the following section), it indicates that the width of the ‘horizon’ in the app, i.e., available activities, is essential.

### 3.2. Healthy Horizons and Healthy Cooperation—The Participants’ Experiences with the mHealth App

A key finding in the analysis of the healthy horizon is that the width of the horizon, i.e., the amount of perceived available health support, is critical to the experience of social support. This complements existing literature (cf. [6]). Some of the first comments from Pilot B had to do with limited activities being available in addition to the group activities. This indicates that participants did browse through the horizon of healthy activities in their local area but experienced a very limited horizon. As one of them stated, “*there is almost nothing*.”

Moreover, among the visible activities, they expressed that activities with a social focus, rather than a simple health focus, most attracted their attention. For example, a participant in Pilot B expressed:

“*I just saw this activity with a bike repair shop, where they mess around downstairs with bikes and repairs and stuff like that. This is the only activity I have found in the app which I actually considered going to*.”(Participant in Pilot B)

Other participants discussed the application’s use of mail-district codes to identify local health activities. In Pilot B, participants discussed whether they would be interested in participating in activities far from their local area and concluded that they were only interested in their immediate residential zone activities. None of them were willing to travel a long way to participate in an activity. The participants in Pilot A were all from the same neighborhood and found the feature with local information particularly important. Thus, both groups supported the design for local horizons and cooperation, and the pilot emphasized that using the app for a group of people who do not live in the same neighborhood is (not surprisingly) typically unsuccessful.

Participants highlighted that a healthy horizon is not enough to ensure success. A facilitator for Pilot B pointed out that for most participants, just showing up for group activities can be a challenge, and therefore, there is a need for a gentle push if the participants themselves should be expected to engage in additional activities.

“*When they were told that they can make something themselves [an activity] in the app, they started blinking their eyes. We worked with a group where it was a pressure just to participate in the lifestyle course, so pushing for further activities is a challenge and maybe too much to ask*.” (Facilitator, Pilot B)

However, the analysis also shows that it did not come to the minds of the facilitators from Pilot B that they, as municipality health team facilitators, could use the mHealth technology to give their participants a gentle push toward a healthy horizon. Their introduction to the mHealth application was done by adding a paper flyer to the general information material handed out to the course participants. A need for special attention or a gentle push from time to time was recognized as they related to general health but not involving the mHealth application itself.

The facilitator from Pilot A expressed how he experienced how half the participants followed information updated in the mHealth group. 

“*Especially if there are any changes. If there is something that is not as it usually is, then it is interesting. And I can also sense that people have seen it—if I have written something incorrect, then they comment on it*.”(Facilitator, Pilot A)

The key finding in the analysis of healthy cooperation is that if cooperation’s happen, they are driven by a social agenda. In cases where health was the primary agenda, which was the motivation for most activities as the app is designed around opportunities to create activities for various exercise types only, participation was low. In contrast, in cases where users were creative and devised other types of activities (often using the category ‘other’ available in the app), participants did not sign up but expressed in interviews that they had seen those activities and found them inspiring. Participants in Pilot B explained the importance of the social character of the activities as follows:

“O*n my part, it [the social] is a determiner. Otherwise, I would not participate—if it were an activity just for me, I would never sign up. I am here for social support*.” (Participant, Pilot B)

“*I also think it is important to focus on social support and experiences especially when you need to build up your basic health condition. It is difficult to build up good health and if you do not have social support to help you, it is really difficult*.”(Participant, Pilot B)

In sum, participants called for a horizon of activities but found the content in the app too limited. Pilot A engaged this issue actively as four participants started organizing activities, contributing content to the app. For participants in Pilot B, this was not the case since they did not live in the same residential area, and according to the facilitator, they did not have the resources to take on further activities. The participants appreciated the social dimensions and ambitions of the mHealth app, but the content was deemed to be too limited, and with too little attention paid to social activities.

## 4. Discussion

Participants in both pilot programs had dominant social and health issues challenging their livelihoods. That was why they had been either invited or referred to participate in municipality-coordinated healthy activities. Our data indicate that it is a challenge to design a mHealth application that both invites this target group to a) produce a window into a thriving social commonality and at the same time b) design one for participants in this social commons to have a mutual interest in cooperating toward a (healthy) horizon. For people with plenty of social interaction during their everyday lives, the need for a social horizon to live healthy is not as prevalent. For people living with social- and health-related challenges, the situation is both unique and daunting, and we learned through this pilot that these target groups do not engage per se by having access to a mobile app but need a facilitator to drive forward technology-mediated social health support. These findings support related work that has raised concern that disadvantaged target groups remain invisible in the development of electronic health systems [15,23]. A consequence of this is a limited acceptance of digital health technology among vulnerable groups of citizens [24]. As such, our results are not surprising but raise questions about the implications. One implication from this and related research is to suggest that digital health interventions should be targeted people with high motivation since people with high motivation and resources are more likely to benefit from technology-supported health. However, a relevant alternative would be to research the needs and barriers of vulnerable groups [24] and actively involve them in design processes. This requires further research on how to understand and provide digital health support in situations where contextual factors such as health problems and social problems influence the use of technology in the engagement of health [25].

Data from interviews with the facilitators—all on the municipality payroll but in different jobs—show a very distinct understanding of their role as facilitators. In Pilot B, one of the facilitators disclosed that she knew there was a need for a much closer follow-up on the course participants, e.g., phoning them from time to time, if they were to have an optimal output. However, simply realizing that this need existed did not encourage the employees to see the mHealth application as a relevant support tool that could provide a gentle push for social interaction among the participants, and therefore, also as a possible tool they could promote and use as part of their courses. The mHealth app remained an opportunity that the participants themselves could decide whether or not to use. Perhaps this happened because the intervention was only a pilot and not a new demand for them doing their job, or perhaps it happened because responsibility for encouraging app use was not specifically allocated. In Pilot A, the facilitator had a different approach to testing the mHealth application among the participants. He saw the need for a gentle push and made it his task to create a group for their activities and post and manage the group’s involvement. As a result, participants in Pilot A were more active using the application. This emphasizes the importance of designing a mHealth app with attention to the target group and focusing on the facilitators as key users and social health mediators through their own use of the app. The results also show that that changing health behavior is not a quick fix and supports research emphasizing that the ambition to design for local community development is a long-term process of infrastructure where technology design must be closely interwoven with support of capacity building in the local community [26,27].

In the case of providing perceived available health support, a gentle push from facilitators toward activities in a ‘healthy horizon’ was observed as important in this pilot study. Only three of the participants pushed themselves toward this horizon, while the main body expressed that this was difficult. In general, this finding is not new. The socio-technical aspects of healthcare systems are complex, and research on digital health design has shown that it is a naive assumption that people will automatically react in specific manners to provided health information. Investigating how to support people in converting technology-mediated health information into self-management strategies is an important and ongoing research focus [28,29].

In general, this study’s findings identified the participants’ appreciation of social support, i.e., that designing for social health support is important. These findings also show how difficult it is to balance a social agenda and a health agenda. The participants in this study found too much focus on health and called for more social activities. This is in line with related work, e.g., with time-bank technology that has explored opportunities to mediate social relations among residents in local communities to share resources. These studies showed that users valued the technology’s ability to mediate social relations and, by this, help make users visible and realize that they can support each other in their daily lives [10]. Though the presented research in this paper was aimed to design a mHealth app supporting social health relations, these findings show that finding ways to foreground the social support for the participants is difficult and that this challenge requires more attention in future research. A way to work with this can be to redesign the mobile app’s information architecture, which is presently set up around exercise activities, and instead focus on how to bring exercise to the background and social activities to the foreground. However, the results primarily highlight the complexity of designing technologies specifically for social health support. Despite this research’s ambition to design a new social support program, the application was perceived by the participants as an app with too much health focus and too little social activity support. This emphasizes the complexity related to understanding mechanisms that contribute to the health benefits from social support and calls for future research in relation to the presented mHealth design and the target group and in general.

## 5. Conclusions

The presented design and pilot of a mHealth application to support health behavior changes toward increased physical activity with social technology has highlighted the complexity related to understanding the social processes that contribute to health benefits. In line with existing research, the study highlights the importance of social support and the complexity related to the mechanisms that must be accounted for when designing software to promote social health support. The present study aimed to design just such an application. Overall, the pilot identified challenges related to the designed application, which was perceived as too health-focused and challenges related to the need for facilitating technology-mediated health support. Continued research is needed—especially to contribute insight into how to target digital support for people who are not engaged in a healthy lifestyle and specifically explore ways to bring social activities to the foreground of health-promoting services.

Though this study is part of a long-term research project, the pilot has limitations related to the number of participants. Further research on a larger scale can contribute insights into nuances of importance for the technical and socio-technical designs around mHealth applications for health behavior changes.

## 6. Patents

The MOVE application is developed as free software and can be downloaded from the Apple App Store or Google Play: https://move.snapp.dk/.

## Figures and Tables

**Figure 1 ijerph-17-08383-f001:**
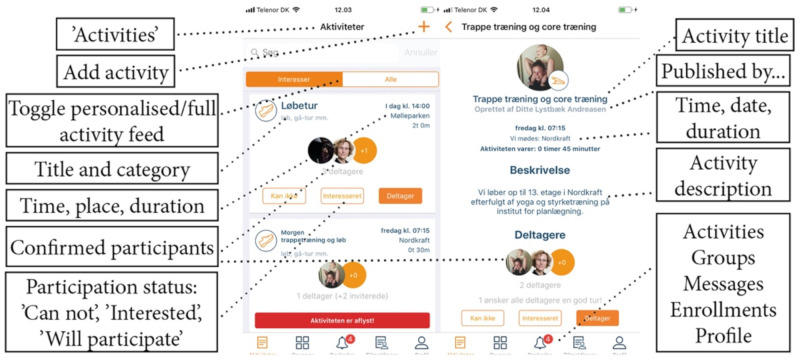
The design of the mobile application. To the left, the entry screen where users can browse through activities in their residential areas (i.e., the healthy horizon). On the right, details and opportunities to join activities (i.e., the ‘healthy cooperation’).

**Figure 2 ijerph-17-08383-f002:**
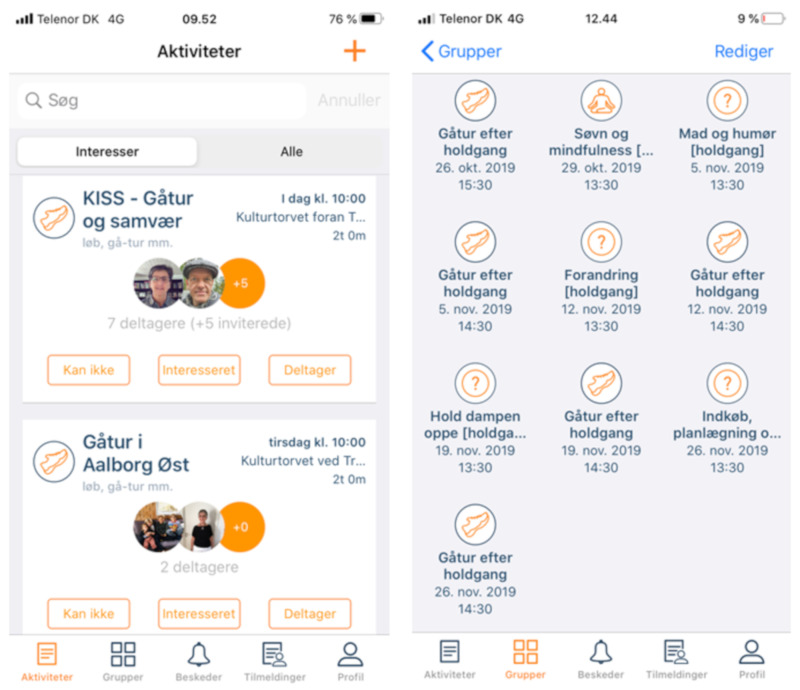
Left: Two activities posted by the facilitator of Pilot A with visualization of members who have signed up for the activity. Right: Activities posted by the facilitator in the Pilot A group.

**Table 1 ijerph-17-08383-t001:** Participants in the pilot tests for the mobile app.

Pilot A	Pilot B	Facilitator
1 male	3 male	Pilot A: 1 community worker (male)
9 female	7 female	Pilot B: Two health professionals (females)

**Table 2 ijerph-17-08383-t002:** Activities and data.

Time	Activity	Data
2 weeks before the pilot	Introductory conversations with the facilitator	NA
Week 0	App set up; a group was set up for Pilot A and Pilot B, and the group activities were created. Introduction to participants	Group interview with participants in Pilot A and Pilot B (n = 20)
Week 1–8 (pilot A) Weeks 1–5 (pilot B)	Pilot/intervention: Facilitators and participants use the app to create and sign up for activities	Online observation of activities in each pilot group
Week 9	Evaluation	Group interview with Pilots A and B (n = 20). Individual interviews with facilitators (n = 3).

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
