# Peer review of "Changing Health Behavior with Social Technology? A Pilot Test of a Mobile App Designed for Social Support of Physical Activity"

_ijerph, 2020, doi:10.3390/ijerph17228383_

Round 1

Reviewer 1 Report

The paper reports on a participatory design experiment where a mobile app aiming at providing social support and engagement for physical activity is evaluated. 

The paper first presents the design of the app (which is available for installation on software stores), details the methods used during the pilots, and finally present the results taken from the tow pilots.

Although the authors acknowledge the limitations of their study, mostly due to the limited number of users involved in the pilots, the authors draw several interesting findings, notably the importance of providing a "health horizon" enabling the users to plan their future activities.

Several minor spelling mistakes in the paper :

.. applications targeted people -> applications target people

participatory designing research -> participatory design research

but challenges -> but challenged

techn ?

our research take place -> our research takes place

digial -> digital

sorrunding -> surrounding

completet -> completed

thje -> the

Author Response

C1: Although the authors acknowledge the limitations of their study, mostly due to the limited number of users involved in the pilots, the authors draw several interesting findings, notably the importance of providing a "health horizon" enabling the users to plan their future activities.

Response: Thank you.

C2: Several minor spelling mistakes in the paper :

.. applications targeted people -> applications target people participatory designing research -> participatory design research but challenges -> but challenged techn ? our research take place -> our research takes place digial -> digital sorrunding -> surrounding completet -> completed thje -> the

Response: Thank you. This is now edited and the paper has been through proofreading. 

Reviewer 2 Report

Although the paper sounds good and overall high quality, the discussion section seems to be not well-organized and not in-depth.

I strongly recommned to reivse the structure and content of the paper, especially analyzing and discussin your research findings.

Author Response

C3: Although the paper sounds good and overall high quality, the discussion section seems to be not well-organized and not in-depth.

Response: The discussion is re-written

C4: I strongly recommend to reivise the structure and content of the paper, especially analyzing and discussion your research findings.

Response: Cf. We have re-strucured and re-written the paper. 

Reviewer 3 Report

This is an excellent manuscript that provides some early feasibility data for the role and importance of social connection and social support in promoting health-positive lifestyle and behavioral changes. As loneliness and social isolation were on the rise pre-COVID and have been accelerated post-COVID, the idea of developing social support-informed interventions to promote healthy behaviors should be at the forefront of population health research. Overall the work is very thoughtfully done and well presented, but I would recommend substantive changes to the structure of the manuscript before it is ready for publication.

Major issues:

  • The structure of the manuscript is unusual and does not appear to follow the typical IJERPH format. The headings of the manuscript should be as follows:
    1. Introduction
    2. Materials and Methods
    3. Results
    4. Discussion
    5. Conclusion
  • The discussion section was seriously lacking in detail and this is the most important issue to address. The authors have made some important observations about the role of social support in promoting health-positive behaviors, but there is some really great literature in this domain that should be discussed in the context of these findings. This will assist the reader in understanding the findings of the manuscript in context. I would recommend a comprehensive re-write of the Discussion section.

Minor issues:

  • There a quite a few typos in the manuscript, so I would recommend a quick sweep of the manuscript to correct some issues. A few examples:
    1. The word "the" misspelled on line 225
    2. The word "mHealth" misspelled on lines 284 and 287
    3. An odd citation method used on line 343
  • I would recommend that app screenshots, such as the ones in Figures 2 and 3, should be translated to english, otherwise they convey very little information. Since this is an English language journal, I think it would be very helpful to perhaps have a second panel in the figure that indicates the translations in English (to be clear: I think the Danish versions of the screenshots should be kept, just translated in the figure). 

Author Response

C5: This is an excellent manuscript that provides some early feasibility data for the role and importance of social connection and social support in promoting health-positive lifestyle and behavioral changes. As loneliness and social isolation were on the rise pre-COVID and have been accelerated post-COVID, the idea of developing social support-informed interventions to promote healthy behaviors should be at the forefront of population health research. Overall the work is very thoughtfully done and well presented, but I would recommend substantive changes to the structure of the manuscript before it is ready for publication..

Response: Thank you.

C6: The structure of the manuscript is unusual and does not appear to follow the typical IJERPH format. The headings of the manuscript should be as follows:
1. Introduction
2. Materials and Methods
3. Results
4. Discussion
5. Conclusion.

Response: The paper is re-structured and now follow the manuscript guidelines with IMRAD structure. 

C7: The discussion section was seriously lacking in detail and this is the most important issue to address. The authors have made some important observations about the role of social support in promoting health-positive behaviors, but there is some really great literature in this domain that should be discussed in the context of these findings. This will assist the reader in understanding the findings of the manuscript in context. I would recommend a comprehensive re-write of the Discussion section..

Response: The discussion is re-written. 

C8: There a quite a few typos in the manuscript...

Response: The manuscript has been proofread.

C9: I would recommend that app screenshots, such as the ones in Figures 2 and 3, should be translated to english, otherwise they convey very little information. Since this is an English language journal, I think it would be very helpful to perhaps have a second panel in the figure that indicates the translations in English (to be clear: I think the Danish versions of the screenshots should be kept, just translated in the figure).

Response: English labels have been insterted where applicable. 

Reviewer 4 Report

The term pilot study should be removed form the title and therm qualitative should replace it. 

Author Response

C10: The term pilot study should be removed form the title and therm qualitative should replace it.

Response: Thank you. The title has been changed. 

Round 2

Reviewer 2 Report

the authors have addressed the issues and questions I raised. 

Author Response

Thank you. We have made further revisions based on good comments from the editing reviewer.